# A First Step to the Categorical Logic of Quantum Programs

**DOI:** 10.3390/e22020144

**Published:** 2020-01-24

**Authors:** Xin Sun, Feifei He

**Affiliations:** 1Department of foundation of computer science, the John Paul II Catholic University of Lublin, 20-950 Lublin, Poland; 2Institute of logic and cognition, Sun Yat-sen University, Guangzhou 510970, China; hliheng@gmail.com

**Keywords:** quantum logic, quantum computing, category theory

## Abstract

The long-term goal of our research is to develop a powerful quantum logic which is useful in the formal verification of quantum programs and protocols. In this paper we introduce the basic idea of our categorical logic of quantum programs (CLQP): It combines the logic of quantum programming (LQP) and categorical quantum mechanics (CQM) such that the advantages of both LQP and CQM are preserved while their disadvantages are overcome. We present the syntax, semantics and proof system of CLQP. As a proof-of-concept, we apply CLQP to verify the correctness of Deutsch’s algorithm and the concealing property of quantum bit commitment.

## 1. Introduction

Quantum programs and quantum protocols are two pillars of quantum computing. The exponential speedup provided by Shor’s factoring algorithm and the quadratic speedup provided by Grover’s search algorithm over their classical counterparts has brought quantum computing into the limelight. The unconditional security offered by quantum protocols such as quantum key distribution has grabbed strong interests from both the academic and the industrial community. Designing quantum programs and protocols is an error-prone task due to the counter-intuitive nature of quantum systems. Therefore, verification techniques for quantum programs and quantum protocols will be indispensable in the coming era of quantum computation. For example, a number of quantum process calculi [1,2,3,4] have been proposed for the formal verification of quantum protocols. Some quantum logics have been developed to verify both quantum programs and quantum protocols.

Quantum logic began with Birkhoff and von Neumann [5]. Traditional quantum logic [5,6,7,8,9,10] focuses on the order-theoretic structure of testable properties in the quantum world and is based on the lattice of closed subspaces of a (usually infinite dimensional) Hilbert space. The success of quantum computation and quantum information has inspired new quantum logics [11,12] which are based on *finite* dimensional Hilbert spaces, such as qubits. Brunet and Jorrand [13] proposed to extend the Birkhoff–von Neumann quantum logic to reasoning about quantum programs. Chadha et al. [14] presented a first attempt to develop a Hoare-like logic for quantum programs. Some useful proof rules for reasoning about quantum programs were introduced by Fenget et al. [15]. A full-fledged Hoare-like logic for both partial and total correctness of quantum programs was established in Ying [16].

The logic for quantum programs (LQP) [17,18,19,20,21] is an extension of traditional quantum logic and quantum Hoare logic. It has been used to verify quantum search algorithms [20], quantum leader election [20], quantum key distribution [21] and quantum voting [22]. The expressive power of LQP is largely determined by the constant symbols it incorporates. There is no systematic study of constant symbols in the literature of LQP. In Baltag and Smets [18], the authors chose the following unitary operators as constant symbols: X, Z, H and CNOT. In Rad et al. [22], Bell states are used as constant symbols. Those operators are not universal. Therefore, there are still many quantum states and operators that cannot be expressed in LQP. Another limitation of the presentation of constant symbols in LQP is the missing satisfying axiomatization. Different axioms for different constant symbols are introduced, depending on which programs/protocols are to be verified. These two limitations make LQP not a convenient tool in the formal verification of quantum programs and protocols: To verify a quantum program or protocol in LQP, we have to first find an appropriate set of constant symbols to express the program/protocol, then we still need to introduce a set of axioms for these constant symbols such that some desired properties of the targeting program/protocol can be proved axiomatically. This procedure usually consumes a lot of time and intelligence. We believe that LQP cannot be successful in formal verification if these two limitations are not overcome.

In this paper, we will overcome these limitations by extending LQP to the categorical logic of quantum programs (CLQP). CLQP is a combination of LQP and categorical quantum mechanics (CQM) [23,24,25,26]. The main feature of the construction of CLQP is the representation of constant symbols of LQP by morphisms in the ZX-calculus, a graphical calculus of CQM. Inherited from the universality of the ZX-calculus [24], CLQP has stronger expressive power than LQP. CLQP also inherits the graphical axiomatization of the ZX-calculus such that many properties of a quantum program and protocols can be proved concisely and graphically, when the program/protocol is specified in CLQP. On the other hand, CLQP preserves the various logical operations (for example, boolean connectives, programs constructors, epistemic modality and probabilistic modality) of LQP. These operations allow us to express various properties that are not expressible in CQM. In a nutshell, CLQP keeps the advantages of both LQP and CQM, while overcoming their limitations. These features make CLQP a powerful tool for the formal specification and verification of quantum programs and protocols.

The structure of this paper is as follows. In Section 2 we will introduce the syntax, semantics and axiomatization of CLQP. Then in Section 3 we apply CLQP to the verification of quantum programs and protocols. We conclude this paper with future work in Section 4.

## 2. Categorical Logic for Quantum Programs

### 2.1. Syntax

For each natural number n≥1, we build the *n*-qubit categorical logic of quantum programs CLQP${}_{n}$. To build the language of CLQP${}_{n}$, we are given the following: A finite set of natural numbers N={1,…,n}, a countable set of propositional variables P, a countable set of operational variables O, constants symbols of propositions and unitary operations built from the ZX-calculus.

#### 2.1.1. Constants from Categorical Quantum Mechanics

Categorical quantum mechanics is the study of quantum computation and quantum foundations using category theory. The ZX-calculus, a graphical language of quantum computation developed in the framework of CQM, is introduced by Coecke and Duncan [24,27]. It is founded on a dagger symmetric monoidal category (†-SMC) C. (A concise introduction to †-SMC is provided in the Appendix A.)

The objects of C are natural numbers: 0,1,2,⋯; the tensor of objects is just the addition of numbers: m⊗n=m+n. 0 is the unit object of C. In the matrix interpretation of the ZX-calculus, an object *n* is interpreted as the $2^{n}$ dimensional Hilbert space $C^{2^{n}}$. An identity morphism of C is interpreted as the identity map on the corresponding Hilbert space. A swap morphism $\sigma_{m,n}$ is interpreted as the map $SWAP_{m,n}$ to which we have $SWAP_{m,n}(|a\rangle |b\rangle )=|b\rangle |a\rangle$ for all $|a\rangle \in C^{2^{m}}$ and $|b\rangle \in C^{2^{n}}$. Apart from the identity morphisms and swap morphisms, the following are also the basic morphisms of C:Z-spiders $Z_{m}^{n}\left(\alpha \right):m\to n$, for every real number α∈R, of which the matrix interpretation and graphical representation are respectively
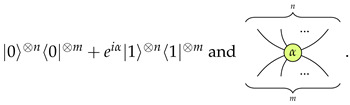

We call α the phase of $Z_{m}^{n}\left(\alpha \right)$. The graph is read from bottom to top. We consider the wires at the bottom as input and those on the top as output. We usually omit the phase in the graphical representation when it is 0.X-spiders $X_{m}^{n}\left(\alpha \right):m\to n$, for every real number α∈R, of which the matrix interpretation and graphical representation are respectively
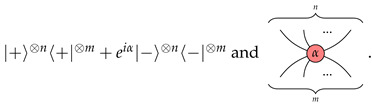

Similarly, we call α the phase of $X_{m}^{n}\left(\alpha \right)$.The identity morphism Id:1→1. The matrix interpretation of Id is the identity map on the Hilbert space $C^{2}$. The graphical representation of Id is a straight line

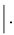
The H-box H:1→1, of which the matrix interpretation and graphical representation are respectively

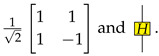
e:0→0 is interpreted as the number 1. The graphical representation of *e* is the empty graph

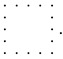
SWAP morphism σ:2→2, of which the matrix interpretation and graphical representation are respectively

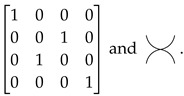
Bell state β:0→2, of which the matrix interpretation and graphical representation are respectively

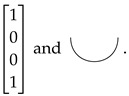
Bell effect β:2→0, of which the matrix interpretation and graphical representation are respectively

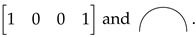


The morphisms of C are generated by applying sequential and parallel composition of the basic morphisms, or by applying the dagger operation † to a morphism. The matrix interpretation of sequential and parallel composition of morphisms is the matrix production and tensor product, respectively. The graphical representation of sequential composition of morphisms is to put one graph on top of another, while the parallel composition of morphisms is represented by putting graphs side-by-side. The matrix interpretation of † is the adjoint operation. The graphical representation of † is to turn the graph upside-down, meanwhile changing the sign of phases that appear in the graph.

The ZX-calculus is a universal language for quantum computing in the sense that it can represent all linear maps between qubit systems. Another impressive feature of the ZX-calculus is that it admits a sound and complete axiomatization [28,29,30] to derive equations between morphisms. The axiomatization of the ZX-calculus will play an important role in the axiomatization of CLQP. There are two kinds of axioms to determine whether two morphisms of C are equivalent: The structure axioms for C as a †-SMC, as well as the rewriting axioms listed in Figure 1. Note that we identify the basic morphism Id with the identity morphism $1_{1}\in C\left(1,1\right)$ and the basic morphism SWAP as the swap morphism $\sigma_{1,1}\in C\left(2,2\right)$. Also note that those axioms listed in Figure 1 are only a fragment of all axioms of the ZX-calculus. We omit other axioms for the sake of simplicity. The interested readers can found all axioms of the ZX-calculus in [29,30].

#### 2.1.2. Syntax of Clqp

We use $L_{n}$ to denote the language of CLQP${}_{n}$. $L_{n}$ is defined by the following BNF:

**Definition** **1**(Language of CLQP${}_{n}$)**.**
*For p∈P and U∈O,*
$a:=U|U^{†}|c_{o}|\phi ?|a;a|a\cup a|a^{*}$$\phi :=⊤|p|c_{p}|\neg \phi |\phi \wedge \phi |\left[\alpha \right]\phi |K_{I}\phi |P^{\geq r}\phi$

Here $c_{o}$ and $c_{p}$ are respectively an operational constant and a propositional constant expressed by the ZX-calculus. More precisely, $c_{o}$ is a morphism from *n* to *n*, while $c_{p}$ is a morphism from 0 to *n*. For example, $Z_{2}^{2}\left(\pi \right)$ is an operational constant of $L_{2}$ and $X_{0}^{3}\left(\pi \right)$ is a propositional constant of $L_{3}$. For all I⊆N and I≠∅, $K_{I}$ is an epistemic modality. For all r∈[0,1], $P^{\geq r}$ is a probabilistic modality.

The intended meaning of those formulas is the following:a is a program.*U* is an operational variable that refers to a unitary operation on $C^{2^{n}}$.$U^{†}$ is the adjoint of *U*.$c_{o}$ is an operational constant that refers to a specific operation on $C^{2^{n}}$.ϕ? is the program that refers to the test of proposition ϕ.$a_{1};a_{2}$ is the sequential composition of $a_{1}$ and $a_{2}$ (applying first $a_{1}$ and then $a_{2}$).$a_{1}\cup a_{2}$ is the non-deterministic choice between of $a_{1}$ and $a_{2}$ (applying either $a_{1}$ or $a_{2}$).$a^{*}$ is the iteration of a, meaning that to repeat a a finite, but non-deterministically determined, number of times.ϕ is a formula.⊤ is a propositional constant representing logical truth.*p* is a propositional variable.$c_{p}$ is a propositional constant that refers to a specific state on $C^{2^{n}}$.¬ is the classical negation.∧ is the classical conjunction.[a]ϕ means that “ϕ will be the case after every execution of a”.$K_{I}\phi$ means that “subsystem *I* carries the information that ϕ is the case”.$P^{\geq r}\phi$ means that “testing property ϕ (on the current state) will succeed with probability ≥r”.

We define logical absurdity as ⊥:=¬⊤, classical disjunction as ϕ∨ψ:=¬(¬ϕ∧¬ψ) and quantum negation as ∼ϕ:=[ϕ?]⊥. Classical implication and equivalence are respectively defined as ϕ→ψ:=¬ϕ∨ψ and ϕ↔ψ:=(ϕ→ψ)∧(ψ→ϕ). Quantum join is defined as ϕ⊔ψ:=∼(∼ϕ∧∼ψ). A range of probabilistic formulas are defined as: $P^{\leq r}\phi :=P^{\geq (1-r)}∼\phi$, $P^{=r}\phi :=P^{\geq r}\phi \wedge P^{\leq r}\phi$, $P^{>r}\phi :=\neg P^{\leq r}\phi$, $P^{<r}\phi :=\neg P^{\geq r}\phi$.

Comparing to LQP, the syntax of CLQP is an extension of LQP with the following additional components: operational constants, propositional constants and the iteration ∗. The iteration ∗ is not included in LQP. We put it back into our logic such that it can be used to verify quantum programs with the while-loop, for example the quantum walk algorithm [31] and the famous HHL (Harrow–Hassidim–Lloyd) quantum algorithms for solving systems of linear equations, which is a cornerstone of quantum machine learning.

### 2.2. Semantics

The semantics of CLQP${}_{n}$ is based on the following structure.

**Definition** **2**(Quantum Dynamic Frame [18])**.**
*Let $H=C^{2^{n}}$ be the $2^{n}$ dimensional Hilbert space. The n-qubit quantum dynamic frame build on H is the following structure:*
$$\Sigma \left(H\right):=(\Sigma ,{\left\{\overset{S?}{\to }\right\}}_{S\in T},{\left\{\overset{U}{\to }\right\}}_{U\in U}).$$
*1.* Σ *is the set of all one-dimensional subspace of H, called the set of states. We denote a state $s=\bar{x}$ of H using any of the non-zero vector x∈H that generates it.**2.* Call two states s and t orthogonal and write s⊥t if and only if ∀x∈s and ∀y∈t, 〈x|y〉=0. For a set of states S⊆Σ, we put $S^{⊥}:=\left\{t\in \Sigma :t⊥s,\forall s\in S\right\}$ and we denote $\bar{S}={\left(S^{⊥}\right)}^{⊥}$ the biorthogonal closure of S.*3.* A set of states S⊆Σ is called a testable property iff it is biorthogonally closed, i.e., $S=\bar{S}$. We denote T⊆P(Σ) the set of all testable properties.*4.* Every testable property S uniquely corresponds to a subspaces $W_{S}$ of H by taking $W_{S}:=⋃S$.*5.* *For every testable property S, there is a partial map S? on *Σ*, called a quantum test, induced by $P_{W_{S}}$ the projector onto the subspace $W_{S}$:*$S?\left(\bar{x}\right):=\bar{P_{W_{S}}\left(x\right)}\in \Sigma$, if $\bar{x}\notin S^{⊥}$.$S?(\bar{x}):=$ undefined, otherwise.*We denote by $\overset{S?}{\to }\subseteq \Sigma \times \Sigma$ the binary relation corresponding to the partial map S?, i.e., given by $s\overset{S?}{\to }t$ iff S?(s)=t.**6.* U is the set of all unitary maps on H. For every unitary map U on H, the corresponding binary relation $\overset{U}{\to }\subseteq \Sigma \times \Sigma$ is given by $s\overset{U}{\to }t$ iff U(x)=y for some vector x∈s,y∈t.

An *n*-qubit quantum dynamic model is a pair M=(Σ(H),V,R), in which $\Sigma \left(H\right):=(\Sigma ,{\left\{\overset{S?}{\to }\right\}}_{S\in T},{\left\{\overset{U}{\to }\right\}}_{U\in U})$ is an *n*-qubit quantum dynamic frame. *V* is valuation function which maps every p∈P to a set of states V(p)⊆Σ and every $c_{p}$ to a singleton set which contains the special state indicated by $c_{p}$. *R* is an interpretation function that maps quantum programs to relations over Σ. An operational variable *U* is interpreted by a relation $R\left(U\right)=\overset{U}{\to }\subseteq \Sigma \times \Sigma$ induced by a unitary map. An operational constant $c_{o}$ is interpreted by the relation $R\left(c_{o}\right)=\overset{c_{o}}{\to }\subseteq \Sigma \times \Sigma$ induced by the unitary map $c_{o}$ indicates. More precisely, $\left(s,t\right)\in R\left(c_{o}\right)$ iff there are x∈s and y∈t such that $\lceil c_{o}\rceil \left(x\right)=y$, where $\left\lceil c_{o}\right\rceil$ is the matrix interpretation of $c_{o}$. A test ϕ? is interpreted by the relation R(ϕ?)⊆Σ×Σ induced by the quantum projector $P_{W_{\bar{V(\phi )}}}$. The adjoint of a program $U^{†}$ is interpreted by the relation $R(U^{†})$, which is defined by $\left(s,t\right)\in R\left(U^{†}\right)$ iff there are x∈s and y∈t such that $U^{†}\left(x\right)=y$. The relational interpretation is extend to arbitrary quantum programs as follows:$R\left(a_{1};a_{2}\right)=R\left(a_{2}\right)\circ R\left(a_{1}\right)$.$R\left(a_{1}\cup a_{2}\right)=R\left(a_{1}\right)\cup R\left(a_{2}\right)$.$R\left(a^{*}\right)={\left(R\left(a\right)\right)}^{*}$, i.e., the reflexive transitive closure of R(a).

**Definition** **3**(Semantics of CLQP${}_{n}$)**.**
*Let M=(Σ(H),V,R) be an n-qubit quantum dynamic model. Let s∈Σ.*
$M,s{⊧}_{n}p$ iff s∈V(p).$M,s{⊧}_{n}c_{p}$ iff $\left\{s\right\}=V\left(c_{p}\right)$.$M,s{⊧}_{n}\neg \phi$ iff not $M,s{⊧}_{n}\phi$.$M,s{⊧}_{n}\phi \wedge \psi$ iff $M,s{⊧}_{n}\phi$ and $M,s{⊧}_{n}\psi$.$M,s{⊧}_{n}\left[a\right]\phi$ iff for all t, if (s,t)∈R(a) then $M,t{⊧}_{n}\phi$.$M,s{⊧}_{n}K_{I}\phi$ iff for all t, if $s{∼}_{I}t$ then $M,t{⊧}_{n}\phi$. Here the relation ${∼}_{I}$ is defined as follows. For all unit vector x∈s and y∈t, let $\rho_{x}=\left|x\rangle \langle x\right|$ and $\rho_{y}=\left|y\rangle \langle y\right|$ be the density operator of x and y respectively. Let $tr_{N\setminus I}$ be the partial trace over the environment N∖I. Then $s{∼}_{I}t$ holds iff $tr_{N\setminus I}\left(\rho_{x}\right)=tr_{N\setminus I}\left(\rho_{y}\right)$.$M,s{⊧}_{n}P^{\geq r}\phi$ iff for all unit vector x∈s, $\langle x|P_{W_{\bar{V(\phi )}}}|x\rangle \geq r$.

The semantics of CLQP${}_{n}$ largely coincides with the semantics of LQP: For all formulas that appear in both LQP and CLQP${}_{n}$, their semantics in CLQP are the same as their semantics in LQP. As usual, by $\Phi {⊧}_{n}\phi$ we mean for all *n*-qubit quantum dynamic model *M* and all state *s* in *M*, if $M,s{⊧}_{n}\psi$ for all ψ∈Φ, then $M,s{⊧}_{n}\phi$. We say that ϕ is valid in CLQP${}_{n}$ if $\emptyset {⊧}_{n}\phi$.

### 2.3. Axiomatization

Now we introduce a sound proof system for CLQP. This proof system is an extension of the proof system of LQP with axioms of the ZX-calculus. It consists of the following axioms and rules:Axioms of dynamic logic:
-All propositional tautologies.-Kripke Axiom: ⊢[a](p→q)→([a]q→[a]q).-PDL1: ⊢[a;b]p↔[b][a]p.-PDL2: ⊢[a∪b]p↔[a]p∧[b]p.-PDL3: $⊢\left[a^{*}\right]p↔\left(p\wedge \left[a\right]\left[a^{*}\right]p\right)$.-PDL4: $⊢\left[a^{*}\right]\left(p\to \left[a\right]p\right)\to \left(p\to \left[a^{*}\right]p\right)$.Axioms of quantum system, in which we use the abbreviations 〈a〉ϕ:=¬[a]¬ϕ, □ϕ:=∼¬ϕ, ⋄ϕ:=¬□¬ϕ.
-Testability Axiom: ⊢□p→[q?]p.-Partial Functionality: ⊢¬[p?]q→[p?]¬q.-Adequacy: ⊢(p∧q)→〈p?〉q.-Repeatability: ⊢(∼∼p↔p)→[p?]p.-Unitary bijectivity 1: $⊢p\to [U;U^{†}]p$.-Unitary bijectivity 2: $⊢p\to [U^{†};U]p$.-Proper Superpositions: ⊢〈a〉□□p→[b]p.-Adjointness: ⊢p→[q?]□〈q?〉⋄p.Axioms of the epistemic modality:
-K: $⊢K_{I}\left(p\to q\right)\to \left(K_{I}p\to K_{I}q\right)$.-T: $⊢K_{I}p\to p$.-S4: $⊢K_{I}p\to K_{I}K_{I}p$.-S5: $⊢\neg K_{I}p\to K_{I}\neg K_{I}p$.Axioms of probability:
-$⊢P^{\geq 0}p$.-$⊢P^{=1}⊤$.-$⊢P^{=0}\left(p↔∼p\right)$.-$⊢\left(p↔q\right)\to \left(P^{=r}p↔P^{=r}q\right)$.-$⊢□□\left(p\to ∼q\right)\to (P^{=r}\left(p⊔q\right)\to \left(P^{=s}p\to P^{r-s}q\right))$.-$⊢\left(□□\left(p\to q\right)\wedge P^{=r}q\wedge \left[q?\right]P^{=s}p\right)\to P^{=rs}p$.-$⊢\left(□□\left(p\to q\right)\wedge P^{>0}p\wedge P^{>0}q\right)\to P^{>0}\left(P^{=r}p\wedge P^{=1-r}q\right)$.Rules:
-Modus Ponens (MP): $\frac{\phi \;\;\phi \to \psi }{\psi }$.-Necessitation (Nec): $\frac{\phi }{[a]\phi }$.-Uniform Substitution (US): $\frac{\phi (p)}{\phi (q/p)}$.-ZX equivalence: if $a_{1}=a_{2}$ in the ZX-calculus, then $⊢\left[a_{1}\right]p↔\left[a_{2}\right]p$.

**Theorem** **1.**
*All axioms and rules above are sound with respect to the semantics of CLQP${}_{n}$, for all n≥1.*


**Proof**.(sketch) Axioms of dynamic logic are valid because they are valid in every dynamic logic and CLQP is a special dynamic logic. The validity of axioms of quantum systems is shown in [17,18]. Axioms of epistemic logic are valid because ${∼}_{I}$ is an equivalence relation. The validity of axioms of probability can be found in [21]. The rules MP and US are valid in all logical systems. The rule Nec is valid because [a] is a necessity modality. The rule ZX equivalence is valid because if $a_{1}=a_{2}$ in the ZX-calculus, then by the soundness of the ZX calculus we know they represent the same linear map. □

**Remark** **1.**
*In CLQP${}_{1}$, axioms of the ZX-calculus are simplified. Those axioms of single-qubit ZX-calculus can be found in Backens [32].*


## 3. CLQP for Verification

In this section we are going to demonstrate the usage of CLQP by applying it to the formal verification of quantum programs and protocols. For the sake of simplicity, we choose to verify the correctness of Deutsch’s algorithm and the concealing property of quantum bit commitment protocols.

### 3.1. Deutsch’S Algorithm

Deutsch’s algorithm is a simple algorithm that solves a slightly contrived problem [33]. It determines whether a function *f* from {0,1} to {0,1} is constant or balanced, where *f* being constant means that f(0)=f(1) and balanced otherwise. We can formalize Deutsch’s algorithm in CLQP${}_{2}$. First, as it is shown in Chapter 12 of [26], we build an oracle $U_{f}$ as the following:

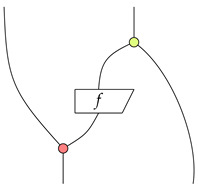

Formally, we have $U_{f}=\left(I⊗Z_{2}^{1}\left(0\right)\right)\circ \left(I⊗f⊗I\right)\circ \left(X_{1}^{2}\left(0\right)⊗I\right)$. It holds that for all f:{|0〉,|1〉}→{|0〉,|1〉}, we have $U_{f}(|x,y\rangle )=|x\rangle |y⊕f\left(x\right)\rangle$. The unitary operation of Deutsch’s Algorithm is graphically represented as the following:

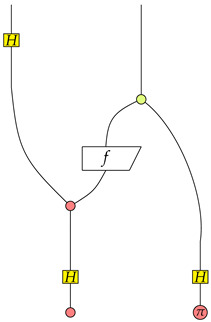

After this operation, a measurement on the {|0〉,|1〉} basis is applied to the first qubit. The function *f* is constant if and only if the result of the measurement is |0〉.

There are four functions from {|0〉,|1〉} to {|0〉,|1〉}. Let them be $f_{1},f_{2},f_{3},f_{4}$, where $f_{1}(|0\rangle )=|0\rangle ,f_{1}(|1\rangle )=|1\rangle$, $f_{2}(|0\rangle )=|1\rangle ,f_{2}(|1\rangle )=|0\rangle$, $f_{3}(|0\rangle )=f_{3}(|1\rangle )=|0\rangle$, $f_{4}(|0\rangle )=f_{4}(|1\rangle )=|1\rangle$. Note that up to a non-zero scalar, we have $f_{1}=X_{1}^{1}\left(0\right)$, $f_{2}=X_{1}^{1}\left(\pi \right)$, $f_{3}=X_{0}^{1}\left(0\right)\circ Z_{1}^{0}\left(\pi \right)$, $f_{4}=X_{0}^{1}\left(\pi \right)\circ Z_{1}^{0}\left(\pi \right)$.

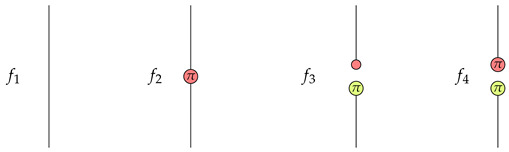

The correctness of Deutsch’s Algorithm can be expressed by the following deduction:$X_{0}^{1}\left(0\right)⊗X_{0}^{1}\left(\pi \right){⊧}_{2}\left[\left(H⊗I\right)\circ U_{f}\circ \left(H⊗H\right)\right]P^{=1}K_{1}\left(X_{0}^{1}\left(0\right)\right)$ for all constant function *f*.$X_{0}^{1}\left(0\right)⊗X_{0}^{1}\left(\pi \right){⊧}_{2}\left[\left(H⊗I\right)\circ U_{f}\circ \left(H⊗H\right)\right]P^{=0}K_{1}\left(X_{0}^{1}\left(0\right)\right)$ for all balanced function f.
Equivalently, it can be characterized by the validity of the following formula of CLQP${}_{2}$:
$$\left(X_{0}^{1}\left(0\right)⊗X_{0}^{1}\left(\pi \right)\right)\to \left(\left[\left(H⊗I\right)\circ \left(U_{f_{1}}\cup U_{f_{2}}\right)\circ \left(H⊗H\right)\right]P^{=1}K_{1}\left(X_{0}^{1}\left(0\right)\right)\right)\wedge ,$$
$$\left(X_{0}^{1}\left(0\right)⊗X_{0}^{1}\left(\pi \right)\right)\to \left(\left[\left(H⊗I\right)\circ \left(U_{f_{3}}\cup U_{f_{4}}\right)\circ \left(H⊗H\right)\right]P^{=0}K_{1}\left(X_{0}^{1}\left(0\right)\right)\right)$$.

The graphical representation of the ZX-calculus can significantly simplify the proof of the validity of the above formula.
Suppose the function is $f_{1}$, then we have

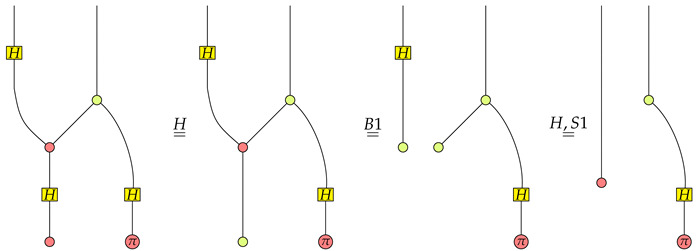

From the rightmost graph we know that $P^{=1}K_{1}\left(X_{0}^{1}\left(0\right)\right)$ is satisfied the first qubit is in the $X_{0}^{1}\left(0\right)$ state.Suppose the function is $f_{2}$, then we have

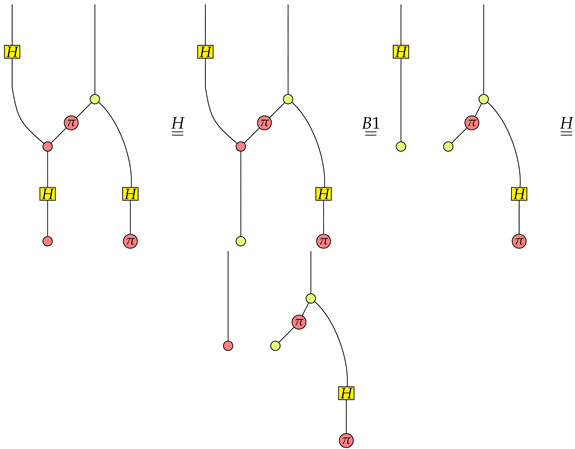

Apparently, $P^{=1}K_{1}\left(X_{0}^{1}\left(0\right)\right)$ is satisfied in the rightmost graph.Suppose the function is $f_{3}$, then we have

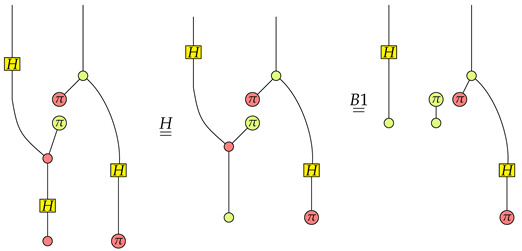

Note that $Z_{1}^{0}\left(\pi \right)\circ Z_{1}^{0}\left(0\right)$ is 0. We know that $P^{=0}K_{1}\left(X_{0}^{1}\left(0\right)\right)$ is satisfied in the rightmost graph.Suppose the function is $f_{4}$, then we have

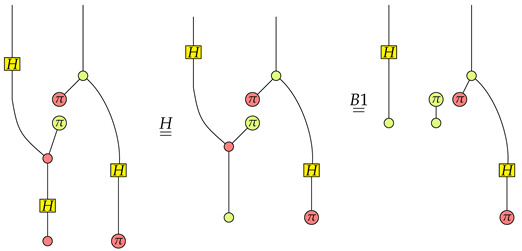

Since the absorbing scalar 0 is in the rightmost graph, we know that $P^{=0}K_{1}\left(X_{0}^{1}\left(0\right)\right)$ is satisfied.

### 3.2. Quantum Bit Commitment

A bit commitment protocol consists of two phases: commit and opening. In the commit phase, Alice the sender chooses a bit *a* (a=0 or 1) which she wishes to commit to the receiver Bob. Then Alice presents Bob some evidence about the bit. The committed bit cannot be known by Bob prior to the opening phase. Later, in the opening phase, Alice announces some information for reconstructing *a*. Bob then reconstructs a bit $a^{\prime }$ using Alice’s evidence and announcement. A correct bit commitment protocol will ensure that $a^{\prime }=a$. A bit commitment protocol is concealing if Bob cannot know the bit Alice committed before the opening phase and it is binding if Alice cannot change the bit she committed after the commit phase.

Quantum bit commitment (QBC) was first proposed by Bennett and Brassard in 1984 [34]. In a QBC protocol, quantum operation and communication are used to ensure the concealing and binding property. In the literature [35,36,37], it is acknowledged that a general model of QBC protocols should at least includes the following ingredients:The Hilbert space required to describe the protocol is the tensor product of the Hilbert spaces that play a role in the protocol.The total system is initially in a pure state.Every action taken by a party corresponds to that party performing a unitary operation on the systems in his/her possession.Every communication corresponds to a party sending a subset of the systems in his/her possession to the other party.
Bearing this common knowledge in mind, a rigorous and simple formalization of quantum bit commitment is given as follows.

**Definition** **4.**
*A quantum bit commitment protocol consists of the following:*
*1.* 
*Two finite dimensional Hilbert spaces A and B.*
*2.* 
*Two pure states |L〉,|R〉∈A⊗B.*
*3.* 
*A completely positive map Open on A⊗B such that Open(|L〉〈L|) is orthogonal to Open(|R〉〈R|).*



This formalization provides a high level description of quantum bit commitment. Initially, Alice (possibly with the help of Bob) prepares a state |L〉 or |R〉 of quantum system A⊗B depending on the value of Alice’s bit. (Note that |L〉 and |R〉 are not the initial state of the QBC protocol, but the final state of the commit phase. Starting from a pure state, a commit phase may involve many rounds of actions and communications). Alice sends $Tr_{A}(|L\rangle \langle L|)$ or $Tr_{A}(|R\rangle \langle R|)$ to Bob to perform the commitment. At the opening stage, Alice sends the rest sub-state of |L〉 or |R〉 to Bob to allow him to verify her commitment. Bob applies the completely positive map Open to determine Alice’s commitment. The QBC protocol is concealing if $Tr_{A}(|L\rangle \langle L|)=Tr_{A}(|R\rangle \langle R|)$. It is binding if there is no unitary map *U* on *A* such that (U⊗I)|L〉=|R〉.

In CLQP, the concealing property of the QBC protocol can be characterized by the validity of the following formula:
$$K_{B}c_{L}↔K_{B}c_{R}$$
Here $c_{L}$ and $c_{R}$ are respectively the propositional constant that characterized the state |L〉 and |R〉. The universality of the ZX calculus ensures that $c_{L}$ and $c_{R}$ can be characterized in CLQP. The validity of this formula implies that $\bar{\left\{|L\rangle \right\}}{∼}_{B}\bar{\left\{|R\rangle \right\}}$, which further entails that $Tr_{A}(|L\rangle \langle L|)=Tr_{A}(|R\rangle \langle R|)$. It seems the binding property cannot be characterized by formulas of CLQP. However, we can still use the ZX-calculus to prove non-binding since the ZX-calculus is universal and sound.

## 4. Conclusions and Future Work

In this paper we introduce the basic ideas of the categorical logic of quantum programs. We present the syntax, semantics and proof system of this logic and demonstrate its usage in the formal verification of quantum programs and protocols. In a nutshell, CLQP is an extension of LQP with a universal set of constant symbols and iteration ∗.

Our long-term goal is to develop CLQP as a powerful tool for the verification of quantum programs and protocols. In the recent future, we will study the decidability and complexity of CLQP. We will also apply CLQP to the formal verification of more complicated quantum programs and protocols, for example the HHL algorithm in quantum machine learning. The semantics of CLQP presented in this paper is based on pure quantum states. The development of mixed-state semantics is in our agenda.

## Figures and Tables

**Figure 1 entropy-22-00144-f001:**
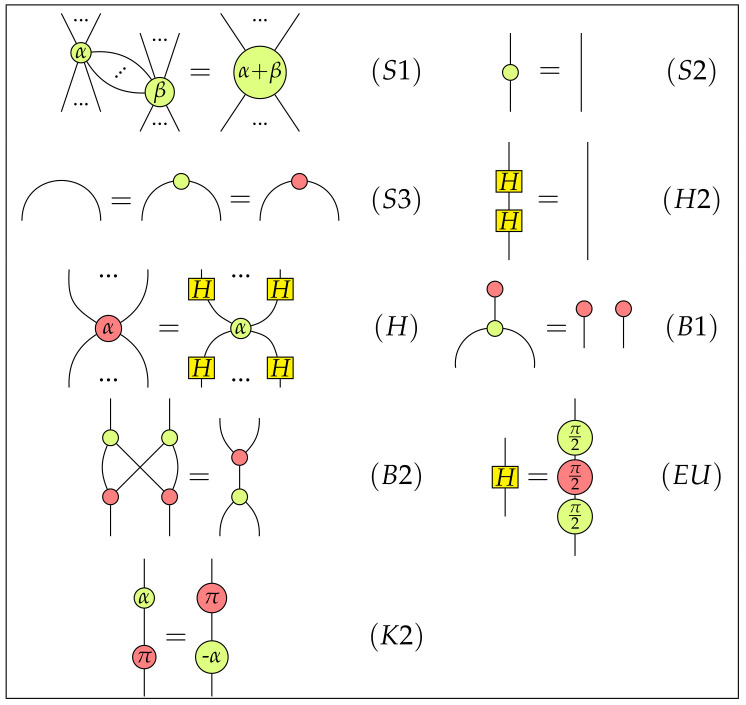
ZX-calculus rules.

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
