# Peer review of "A First Step to the Categorical Logic of Quantum Programs"

_entropy, 2020, doi:10.3390/e22020144_

Round 1
Reviewer 1 Report
In this paper, the acronym CLQP is used for categorical LQP (the logic of quantum programs) and CLQP$_n$ is for $n$ qubits.
The starting idea of categorical QM (quantum mechanics) is due to Coecke and Duncan [24] and the present 'first step' of CLQP owes much to the recent development
of ZX calculus. [28-30].
The goal of the paper is to provide the necessary technical details for the synthax, semantics and proof system of CLQP. It is applied to Deutsch's algorithm that works in CLQP$_2$ and to quantum bit commitment.
The paper is interesting and well written. A few remarks
* In definition 1, $\alpha$ has not the same meaning than in the Z-spider $Z_m^n(\alpha)$.
* Could you justify the equation in line 308?
A few misprints:
l 24: attempt; l 33: are used; l 63: in the correspondence; l 295: Hilbert; l 305: concealing.
Author Response
We thank the review for her/his valuable comments. Please find our reply attached.

Reviewer 2 Report
The paper describes a combination of LQP and the ZX-calculus. LQP (Logic of Quantum Programs) refers to an existing dynamic quantum logic. The ZX-calculus is a pictorial calculus for reasoning about equalities of finite-dimensional linear operators. (The authors call it a combination of LQP and Categorical Quantum Mechanics, but in fact they just use the specific calculus ZX.)
The logic LQP has is a logic that allows to reason about simple quantum programs and quantum predicates. It is parametric in a set of unitary operations that the programs and predicates can use. The authors of the present paper point out that no fixed set of unitaries has been used in papers using LQP so far. Thus, they fix the set of unitaries to be diagrams in the ZX-calculus. Since the ZX-calculus is complete, this means that any unitary can be represented.
I am not sure whether there are additional changes made to LQP because the resulting logic CLQP is given monolithically, i.e., without a discussion what has changed and what not and what concept is from which of the prior work. (I compared with [18], it seems the present paper additionally has the operator K_I on predicates, but maybe that one comes from another paper. At least, there is no mention of its novelty.)
I think the contribution of this paper is very incremental: LQP is already parametric in the set of unitaries employed and leaves it open how they are represented. So all that one needs to do is to say that the set of unitaries are the represented using diagrams from the ZX-calculus, and we are done. I don't recognize what additional benefit is gained from writing the result as one single logic (with ZX hardcoded), in fact, one loses modularity this way.
The only theorem is the fact that the resulting calculus is sound (unsurprisingly, since LQP was). Completeness is not considered, even though the completeness of ZX is mentioned as a reason for considering ZX.
Two examples are given: One is the analysis of Deutsch's algorithm, one is the definition of the hiding property of a bit commitment scheme. Deutsch's algorithm can as easily be analyzed in LQP, basically the example boils down to an illustration how the resulting calculations can be done pictorially (in the ZX-calculus). The second example seems unrelated to the contribution of the paper because it simply defines a predicate that is supposedly representing the hiding property of a commitment scheme, the ZX-calculus does not enter the picture here.
Further comments:
In the description of CLQP, please describe what is new and what is from which prior work. (Only the quantum frame has an explicit citation [18]) In Section 3.2, you define the hiding property of a bit commitment. This is usually defined as (in the language of your paper) |L> \sim_B |R>. Is the CQLP formula you give at the end of 3.2 equivalent to this? If not, why use a nonstandard security definition? In Section 3.2, the binding property of commitments is not formalized. And no example is given.
Author Response
We thank the review for his valuable comments. Please find our reply attached.

Round 2
Reviewer 2 Report
The authors response and revised version addresses most of my minor concerns but the main concern remains: The main contribution of the paper, namely that combining LQP and the ZX-calculus is straightforward. The authors argue in their response (and in the paper) that it is important to have a complete set of unitary operation symbols available when using LQP. I do not disagree with that claim! However, LQP is parametric in the set of unitary operation symbols. That is, when using LQP, one can choose whichever set of operation symbols one wishes to use. Depending on the situation, this could be the set of, say, the set of all Clifford gates, or the set of all matrices with rational entries, or whatever. For whichever set of unitary operation symbols one chooses, one needs to figure out a procedure for proving the equality of two unitary operations. Again, I agree with the authors that this can "consume a lot of time and intelligence", depending on the set of unitary operation symbols that is chosen. But if one chooses a set of operation symbols (e.g., the operations expressible in the ZX calculus) for which there already exists such a procedure (and software tools), then this problem does not apply. One simply uses the existing produce for any equality proofs that arise. In particular, there is no need to restate LQP with the new set of operation symbols (because LQP is parametric), neither is there a need to prove the soundness of the resulting combination of LQP and ZX (because the soundness of LQP holds if our procedure for deciding equality of unitaries is sound, and that procedure is sound because it has been proven sound in the literature about the ZX calculus).
So the main contribution of the paper could be achieved using the following text: "LQP is parametric in the set of unitary operation symbols and predicate constant symbols. We suggest to use the ZX-calculus for these. Then verification conditions involving those symbols can be decided with existing methods from the ZX calculus." Note: I do not mean that this would be an abstract, or a plan for the proof. I claim that this text fragment would be a complete description of LQP+ZX (except possibly for illustrating examples). Stated like this, it becomes clear that the main contribution of the paper does not go beyond a short observation and thus does not, I believe, reach the bar for acceptance in Entropy.
Why is the paper so long if the main contribution could be formulated in three sentences? Mostly, because the logic LQP is spelled out in its entirety when the ZX-calculus is hardcoded into the logic. However, there is no need for that (as stated above) because LQP is already parametric, there is no need to hardcode the ZX-calculus (in fact, doing so only removes flexibility). Secondly, LQP itself is modified: the "iteration \ast" is added to support while loops. However, this change is orthogonal to the main contribution (of adding ZX) and neither necessary for using ZX, not relying on ZX. (I don't know whether this contribution on its own, without the ZX-addition, would be a worthwhile contribution since I don't know how while loops were handled in prior versions of LQP. However, the authors didn't even point out this contribution in the first revision of the paper.)